# Surface Modifications of Wheat Cultivar Bologna upon Treatment with Non-Equilibrium Gaseous Plasma

**DOI:** 10.3390/plants11121552

**Published:** 2022-06-11

**Authors:** Matej Holc, Alenka Vesel, Rok Zaplotnik, Domen Paul, Gregor Primc, Miran Mozetič, Peter Gselman, Nina Recek

**Affiliations:** 1Department of Surface Engineering, Jožef Stefan Institute, Jamova Cesta 39, 1000 Ljubljana, Slovenia; matej.holc@ijs.si (M.H.); alenka.vesel@ijs.si (A.V.); rok.zaplotnik@ijs.si (R.Z.); domen.paul@ijs.si (D.P.); gregor.primc@ijs.si (G.P.); miran.mozetic@ijs.si (M.M.); 2Jožef Stefan International Postgraduate School, Jamova Cesta 39, 1000 Ljubljana, Slovenia; 3Interkorn Ltd., Gančani 94, 9231 Beltinci, Slovenia; peter.gselman@interkorn.si

**Keywords:** wheat, seed, plasma, wettability, surface modification

## Abstract

Seeds of wheat cultivar Bologna were treated with a low-pressure, inductively coupled, radio frequency oxygen plasma. E-mode and H-mode plasma at the real powers of 25 and 275 W, respectively, was used at treatment times of 0.1–300 s. Plasma affected seed surface chemistry, determined by XPS, and surface topography, visualized by SEM. The combined effects of functionalization and etching modified seed surface wettability. The water contact angle (WCA) exponentially decreased with treatment time and correlated with the product of discharge power and treatment time well. Super-hydrophilicity was seen at a few 1000 Ws, and the necessary condition was over 35 at.% of surface oxygen. Wettability also correlated well with O-atom dose, where super-hydrophilicity was seen at 10^24^–10^25^ m^−2^. A relatively high germination percentage was seen, up to 1000 Ws (O-atom dose 10^23^–10^24^ m^−2^), while seed viability remained unaffected only up to about 100 Ws. Extensively long treatments decreased germination percentage and viability.

## 1. Introduction

Wheat is an abundantly produced grain crop that has been a staple of the human diet for thousands of years. To this day, wheat production keeps increasing; since the beginning of the 1960s, its world production has multiplied approximately 3.5 times, reaching a total of nearly 766 million tonnes in 2019. This makes wheat the third most-produced crop worldwide, after sugar cane and maize [1]. In order to facilitate this ongoing production growth and tackle the expanding challenges of climate change, innovative approaches to crop production are sought after. One such area of innovation is the field of plasma agriculture, in which low-temperature plasma technologies are being employed to improve seed and crop properties [2,3].

With wheat, the structure of interest is the wheat kernel, which facilitates the plant’s reproduction and is simultaneously the edible part of the plant used in the human diet. The kernel is a sort of fruit, also called a caryopsis [4]. In practice, it plays the same role as a seed, and thus for the sake of simplicity, we refer to wheat kernels as “seeds” throughout this paper.

During plasma treatment of any material, the majority of the interactions with plasma occur at the very surface of the material. In the case of wheat seeds, the outermost botanical layer is the pericarp, which surrounds the seed coat [4]. The outer layer of the pericarp is the epidermis, of which the outer cell wall is covered with a waxy cuticle [5]. Thus, the cuticle is the first wheat seed structure to interact with the plasma and its reactive species. In wheat seeds, the cuticle is a complex mixture of alkanes, fatty acids, alcohols, sterols, amyrines, and β-diketones. In different cultivars, their content varies between 40 and 60 μg per g of seed [6].

Thus far, wheat is one of the most common agricultural seeds treated in plasma systems for agricultural purposes. Around 50 publications detailing direct plasma treatment of wheat seeds are currently available. The bulk of this research has been published since 2017, but some papers date as early as 2008. The publications explore a wide scope of wheat seed and plant improvement aspects. However, we have not found any crucial correlations between treatment parameters, surface chemistry, germination, and viability by surveying the scientific literature. These correlations are needed in any attempt to understand the complex mechanisms and upscale the experiments for a feasibility study. Our study is the first that reports the correlations between the treatment parameters and the biological response.

Plasma was commonly found to modify the physico-chemical properties of the wheat seed surface. Morphology was mainly investigated using scanning electron microscopy (SEM). Etching of the wheat seed surface was seen after low-pressure capacitively coupled (CC) radio frequency (RF) air plasma treatment [7], medium-pressure glow discharge treatment [8,9], as well as dielectric barrier discharge (DBD) treatment using various gases at atmospheric pressure [10]. By using a glow discharge in argon, the etching of the surface was found to increase with increased treatment time and input power [11]. Further, in an inductively coupled (IC) RF plasma system, etching depended on the direct or indirect nature of treatment; nanostructuring was only seen following glow, but not afterglow, plasma treatment [12].

Further, as widely observed with organic materials, including polymers [13], plasma is known to modify surface chemistry. When oxygen is present in the processing gas, new oxygen-containing functional groups become introduced into the predominantly hydrocarbon structure of the cuticle. Using X-ray photoelectron spectroscopy (XPS), this was shown in wheat seeds treated by atmospheric-pressure DBDs using argon [14] or helium [15] as processing gases. Time-of-flight secondary ion mass spectrometry also indicated the presence of new O-rich functional groups in wheat seeds after low-pressure IC RF air plasma treatment [16]. Other techniques showing plasma-mediated chemical changes on wheat seed surfaces include energy-dispersive X-ray spectroscopy (EDX) and Fourier-transform infrared (FTIR) spectroscopy [15,17]. In order to monitor the plasma chemistry itself, analytical techniques such as OES may be useful to apply during treatment [15,18].

The modified physicochemical properties of the wheat seed surface affect its wettability. A decreased water contact angle (WCA) was measured after a variety of plasma treatments, including DBDs [14,15,19,20], a glow discharge in argon [11], and an IC RF discharge [16]. One group used ethylene glycol and diiodomethane contact angles to further assess the hydrophilicity of the wheat seed’s surface [20]. In relation to increased wettability, water uptake of the seeds commonly increased as well [10,14,15,16,21].

While the exact reasons are still being elucidated, plasma has consistently been shown also to affect seed germination. In wheat seeds, many of the aforementioned plasma treatments also increased the germination rate (GR), including atmospheric pressure DBDs [10,22,23], glow discharges [8], plasma jets [18], and IC RF plasmas [16]. Particularly in commercial varieties, wheat seed germination may already be reasonably high, even close to 100% [15,19,21,24]. In these cases, improvement in germination may add up to no more than a few percent. Further, the germination percentage may decline with prolonged treatment by gaseous plasma due to damage to the seed [19,21]. Naturally, the plasma effect on germination typically depends on the specifics of the plasma system and treatment, such as choice of the process gas [10], treatment duration [20], or choice of either direct or indirect treatment [12,23]. Different wheat cultivars have also been shown to respond differently to the same treatment conditions [12].

In the following experiments, we worked with the wheat cultivar, Bologna. It is a common Italian winter wheat cultivar with hard kernels, red grain color, a thousand kernel weight of 30–35 g [25], and is commonly used in bread making. We have examined the effects of a powerful IC radio frequency RF plasma system on the properties of its seeds. Specifically, we recorded changes to surface wettability and water uptake, as well as surface morphology, due to the plasma exposure. Further, we followed wheat germination and seed viability in a laboratory setting.

## 2. Materials and Methods

### 2.1. Seed Material

Seeds of wheat cultivar Bologna were obtained from Interkorn Ltd. (Gančani, Slovenia). The seeds were kept in a refrigerator at 8 °C in a dry atmosphere to prevent early germination during storage.

### 2.2. Plasma Reactor

Wheat seeds of the cultivar Bologna were exposed to oxygen (O_2_) plasma. The plasma is created in a glass discharge tube with a diameter of 4 cm and a length of about 80 cm. A relatively uniform glow discharge forms within an RF coil that is 15 cm long. The coil is connected to an RF generator operating at a frequency of 13.56 MHz and adjustable output power of up to about 1 kW. The RF generator was type CESAR 1310 (Advanced Energy, Denver, CO, USA). The generator was connected to a Gamma-type matching network (ICP VarioMatch, Advanced Energy, Denver, CO, USA). The matching network consists of two vacuum capacitors of adjustable capacitance. One capacitor is connected in series between the RF generator and the coil, and another in parallel. The role of the matching network is to adjust the impedance of the load to the generator impedance and thus enable the efficient transfer of the generator power into the plasma coil. The experiments were performed at a fixed forward power of 50 or 300 W to achieve E or H-mode plasma, respectively. Details about both modes obtained in oxygen plasma at similar conditions as in this study are explained elsewhere [26]. A schematic representation of the system is seen in Figure 1. Commercially available oxygen with a purity of 99.999% is introduced into the glass tube on one side, whereas the other side is continuously pumped by a two-stage rotary pump. Continuous pumping allows for rapid removal of any reaction products that might otherwise accumulate in the plasma reactor and alter the original gas composition.

### 2.3. Plasma Treatment

During each treatment, multiple wheat seeds were treated simultaneously. The dry seeds were placed into a concave recess of a glass holder, which was placed in the middle of the copper coil, as shown in Figure 1. The E-mode plasma is fairly uniformly distributed along the discharge tube, while the H-mode plasma is mostly spatially enclosed within the coil [26]. In both cases, the seeds were directly exposed to the glowing plasma. The discharges were ignited at a pressure of 25 Pa. The output power at the generator was 50 W for the E-mode and 300 W for the H-mode. The reflected power was 25 W for both modes, so the real power absorbed by the plasma was 25 W and 275 W for the E and H-mode, respectively. Treatment times were 0, 3, 10, 30, 100, and 300 s for the E-mode and 0, 0.3, 1, 3, and 10 s for the H-mode. The times of 300 s and 10 s were the longest suitable treatment times for the E and H-mode, respectively. In both cases, longer treatment times caused visible damage to the seeds during preliminary experiments and were thus not considered for inclusion.

### 2.4. Fluence of Neutral Oxygen Atoms

The major reactants in low-pressure oxygen plasma are neutral oxygen atoms in the ground state because their density is orders of magnitude larger than the density of other reactants, such as oxygen ions. The density of neutral oxygen atoms was measured with a catalytic probe. We used a laser-heated fiber-optics catalytic probe (Plasmadis Ltd., Ljubljana, Slovenia). The accuracy of this technique is about 20% when operating in E-mode plasma and about 40% in H-mode. The increased uncertainty of the probe when plasma is in the H-mode is due to stray effects caused by oxygen ions and vacuum ultraviolet radiation [28]. The measured values are 3.45 × 10^19^ and 4 × 10^21^ m^−3^ for the E and H-modes, respectively. The flux of oxygen atoms on the surface of any object exposed to oxygen plasma is calculated from the standard equation *j* = ¼ *n<v>* where *n* is the density and *<v>* is the average random velocity of O-atoms, i.e., $<v>=\;\sqrt{8kT/\pi m}$. The fluence doses of O-atoms are simply a product of the flux and the treatment time. 

It is worth mentioning that neutral oxygen atoms are not the only reactive species of high density in plasmas sustained by electrodeless high-frequency discharges. Oxygen molecules are excited upon inelastic collisions with plasma electrons to form metastable states of neutral molecules. A particularly large density was reported for O_2_ molecules in the singlet state of the radiative lifetime of almost one hour [29]. These molecules of potential energy just below 1 eV probably interact chemically with organic matter, such as proteins [30]. Unfortunately, the interaction kinetics of singlet oxygen with seeds surface is yet to be explored. 

### 2.5. X-ray Photoelectron Spectroscopy

The elemental composition of the surface layer of seeds, expressed in atomic percent (at.%), was determined by XPS. After wheat seeds were fixed and placed on the sample holder, they were left inside the XPS fore-chamber to dry for 30 min to ensure an appropriate evacuation of the XPS fore-chamber. The analyzed area of the sample was 400 μm^2^, excited by X-ray radiation from a monochromatic Al source at a photon energy of 1486.6 eV and a take-off angle of 45°. XPS survey-scan spectra were acquired at a pass-energy of 187 eV using an energy step of 0.4 eV. Spectra were analyzed using the MultiPak V8.0 software (Ulvac-phi, Inc., Chigasaki, Japan). All measurements were performed in duplicate, and a mean value and standard errors were calculated. The probing depth of XPS for this type of material is several nm.

### 2.6. Scanning Electron Microscopy

The SEM images of wheat seed surface were obtained using a Quanta 650 environmental SEM (Thermo Fisher, Waltham, MA, USA) at 2000× and 5000× magnifications. Environmental SEM enables scanning without the additional coating of the sample surface, even at high magnifications. Prior to imaging, the seeds were attached to an aluminum holder using conductive carbon tape.

### 2.7. Water Contact Angle

WCA was measured on untreated and plasma-treated wheat seeds to determine surface wettability. Static WCA by the sessile drop method was evaluated using the Drop Shape Analyzer DSA 100 (Krüss GmbH, Hamburg, Germany). A droplet of deionized (DI) water with a volume of approximately V = 1 μL was placed onto the wheat seed surface. The measurements were performed under ambient conditions. WCA was measured immediately after plasma treatment. All measurements on each sample were repeated, while the sample at each treatment condition was prepared in triplicate. The mean WCA from three equally treated seeds was calculated together with the standard error.

### 2.8. Germination

The germination of wheat seeds was monitored in laboratory conditions. Each sample (plasma treated and untreated) contained 100 seeds. The number of seeds in the samples was split in half for the germination test, meaning there were 2 parallels of 50 equally treated seeds. A sample of 50 seeds (untreated and plasma-treated) was placed in a Petri dish with a wet filter paper placed at the bottom. On days 4 and 8, the seeds were checked for germination. During 8 days of incubation, tap water was added as needed. The amount of water was kept approximately the same across all Petri dishes to provide and maintain the same conditions for all samples. Enough water was added only to wet the filter paper. The germination percentage of the sample was calculated from the number of germinated seeds.

### 2.9. Viability of Seeds

Seed viability was tested using tetrazolium chloride. After plasma treatment, seeds were halved and put into a Petri dish containing filter paper wetted with a 0.1% solution of triphenyl tetrazolium chloride. After exposure to the chemical, viable embryos of seeds turned to red or pink color due to the reduction of the colorless triphenyl tetrazolium chloride into a soluble colored dye called triphenyl formazone. Seeds that did not turn red or pink were, therefore, not viable. After 24 h, the reaction was completed, and the viable vs. non-viable seeds were counted. Each sample contained 30 seeds and was observed in 2 parallels, from which the mean was calculated together with the standard error.

### 2.10. Statistical Analysis

Data from the WCA, germination, viability, and XPS measurements were statistically analyzed using the JASP 0.9.2. open-source software (University of Amsterdam, Amsterdam, The Netherlands). Group means were calculated and compared using ANOVA followed by the post hoc Tukey’s range test. Differences in the means were considered statistically significant at (*p* < 0.05). Where error bars are shown in charts, they represent the standard error.

## 3. Results and Discussion

### 3.1. X-ray Photoelectron Spectroscopy

The elemental composition of the outermost layer of Bologna wheat seeds for the three most important elements is shown in Figure 2. With increasing treatment time, both in E and H-modes, the concentration of carbon decreases. At the same time, the concentration of oxygen increases to the point where it reaches a plateau, a saturation of the surface with oxygen functional groups. The nitrogen concentration is low. For the E-mode, the C decrease and O increase become statistically significant after treatments of 100 s and longer. For the H-mode, both differences are significant after treatments of 1 s and longer. There does not seem to be a trend in the nitrogen concentration fluctuations.

XPS survey spectra of untreated and treated seeds are presented in Figure 3. The spectra of untreated seeds reveal only C, O, and traces of Si. After plasma treatment, some additional peaks appear, indicating the presence of N and Ca. These elements are present in minor amounts and represent less than 1 at.%. The enhancement of the concentration of trace elements, as revealed from the survey spectra, is explained by the etching of the uppermost seed layer. The microelements are not supposed to be removed from the seed surface by treatment with oxygen plasma, so their peaks in the XPS survey spectra are the highest at the longest treatment times.

Discharges in oxygen-containing gases are well-known to introduce oxygen-containing functional groups to the sample surface. This has been previously shown in seeds of a variety of plant species [16,27,31,32]. In wheat seeds, however, only two previous publications used XPS to analyze surface composition. One group [15] treated wheat seeds with an atmospheric-pressure DBD using helium as the processing gas. During 15 min, the carbon concentration in the surface film gradually decreased from over 90 at.% to about 60 at.%, while the oxygen concentration increased from about 7 at.% to 35 at.%. While these values are comparable to our own, as seen in Figure 2, the authors [15] required a sufficiently longer treatment time due to the differences in the plasma systems (about a minute). Another group [14] also showed an increased proportion of O-containing functional groups on the seed surface after treatment with an atmospheric-pressure DBD using argon. The O/C ratio more than doubled after 60 s of plasma treatment. In the study, the plasma treatment system was not tightly closed, and ambient air was allowed to enter the reactor. The formation of oxygen-rich functional groups on a sample surface when using oxygen-free gas (helium or argon) is also explained by various additional effects. Although the purity of the source gas is high, the desorption of water from the nearby surfaces (including the samples) causes water vapor to enter the discharge. The water vapor is immediately dissociated in the plasma of noble gas and forms OH radicals, which are among the reactants of the highest oxidation potential. The plasma sustained in a noble gas is also a rich source of vacuum ultraviolet (VUV) radiation. The energetic photons cause bond scission, and the dangling bonds are occupied by oxygen after exposure to air [33].

Other publications used different analytical methods to assess the chemistry of wheat seed surface after plasma exposure. After treatment with a low-pressure IC RF air discharge, mass peaks of oxygen determined by time-of-flight secondary ion mass spectrometry (ToF-SIMS) were 2.5–3 times more intense, which also signifies the incorporation of oxygen-containing functional groups [16]. Further, the XPS findings after the atmospheric-pressure DBD with helium were complemented by results of Fourier-transform infrared spectroscopy (FTIR) and energy-dispersive X-ray spectroscopy (EDX). FTIR confirmed the etching of the carbon-containing hydrophobic outer layer, while EDX confirmed that chemical changes were mostly limited to the outer seed layers, and no changes occurred in the seed bulk [15]. FTIR and EDX results were also obtained after DBD treatment using dry air. After plasma exposure, the O/C ratio increased, and other chemical changes were detected by FTIR [17].

### 3.2. Scanning Electron Microscopy

The morphology of Bologna wheat seeds was checked before and after plasma treatments to evaluate the topological changes plasma may cause on the surface of these seeds. SEM images are seen in Figure 4. The surface of untreated wheat seed is wavy, uneven, and full of deep ditches and grooves throughout the whole area (Figure 4a,b). After 300 s treatment in E-mode plasma, where the electron density is much lower than in the H-mode, the surface of a wheat seed seems flatter, and the ditches and grooves are less pronounced. Even more, it appears as if some layers were removed from the surface, which became flattened. After treatment in H-mode, where the electron density inside the coil is very high, etching of the surface is evident. The long linear grooves become eroded and deepened compared to untreated seeds (Figure 4e,f).

Our IC RF plasma system, operated in oxygen at different discharge parameters, was previously shown to etch and nanostructure wheat seed surface in the direct treatment (glow), but not the indirect one (afterglow). However, the used discharge powers were significantly higher, 200 and 600 W for the E and H-modes, respectively [12]. One other work using IC RF plasma, but in air, left the seed surface unchanged; however, after only 15 s of treatment at a low discharge power (20 W), where the plasma was in the E-mode [16]. In another work, native structures similar to those in Figure 4a were extensively etched after 5 min of CC RF air plasma treatment [7]. Medium-pressure glow discharges also resulted in significant etching [8,9]. In one atmospheric-pressure DBD, the extent of etching depended on the process gas and treatment time. In the longest treatments with air plasma (16 and 19 min), the longitudinal grooves of the surface were nearly eroded [10]. 

Plasma etching of the sample surface was carried out by energetic ions that strike the surface and physically erode it [34]. Further, extensive functionalization can remove material from the sample surface by the process of chemical etching [35]. The etching of the surface typically increases with increased treatment time and generator power, as was seen in wheat seeds treated with a glow discharge in argon [11]. The extent of etching seen in Figure 4 is moderate but evident. The reason may lie in the relatively low generator powers used in the experiments.

### 3.3. Water Contact Angle

Immediately after plasma treatments, WCAs were measured for untreated and plasma-treated wheat seeds. The results are shown in Figure 5. The initial static WCA of wheat seeds was about 110°. After plasma treatment in E and H-modes, the WCA decreased significantly. In E-mode, at lower powers of the RF generator and lower electron density [36], the WCA significantly dropped after 3 s of treatment and dropped even further after 10 s. However, the surface becomes highly hydrophilic (WCA about 10°) only after a longer exposure time of 100 s, while after 300 s, it even becomes super-hydrophilic (immeasurably low WCA). Contrarily, in the H-mode, at a higher generator power and higher electron density [36], the surface wettability changes appeared after much shorter treatment times. In H-mode, it only took as little as 0.1 and 0.3 s for the WCA to significantly drop from 110° to 70°. After 10 s, the surface was already super-hydrophilic. 

The results summarized in Figure 5 reveal large differences in the evolution of the surface wettability. While the treatment in weakly ionized plasma (when sustained in the E-mode) causes only a moderate wettability increase even for the treatment time as long as 30 s, the fully wettable surface is achieved within 10 s after the treatment in H-mode plasma. The difference in WCA is statistically significant in all instances except for treatments of 10 and 30 s in the E-mode and is explained by the large differences in the discharge power, which, in turn, affects the plasma parameters [36]. A more decisive parameter than treatment time alone may be the product of the discharge power and the treatment time. Figure 6 reveals the evolution of the surface wettability versus the product of real power and treatment time. The measured points are somewhat scattered, but the general trend is obvious: the static WCA depends monotonously on the product of the treatment time and the RF power absorbed by the gaseous plasma. This trend is explained by the fact that the surface modification of any material does not depend directly on the discharge power but rather on the doses of reactive plasma species. The doses should be a product of the discharge power and the treatment time. From Figure 6, it is revealed that the product of the order of 10 Ws causes a moderate increase in the wheat seed wettability, but the super-hydrophilic surface finish is only achievable as the product of several 100 Ws. These values are specific to the plasma reactor shown in Figure 1.

The reactive particles found in inductively coupled RF plasma are positively and negatively charged molecular and atomic ions, neutral molecules in metastable states, and neutral atoms in the ground and metastable states [37]. Plasma is also a source of radiation, in particular, the radiation in the vacuum UV part of the spectrum [38]. The exact mechanisms of the interaction of all these reactive plasma species with organic materials are still dim. The interaction depends on the fluxes or fluences (doses) of all these species, and synergies are likely to play a role too. Still, the major reactants in the plasma sustained in glass tubes, as in our case, are neutral oxygen atoms in the ground state [39]. This is because, in such systems, the O-atom density is orders of magnitude larger than the density of charged particles. The decisive plasma parameter governing the wettability of any organic material, including wheat seeds, should be the dose of neutral oxygen atoms. 

Figure 7 shows the WCAs measured after plasma treatment in E and H-modes versus the O-atom fluence. The measured points for E and H-modes do not overlap, but the WCA is always larger for the case of plasma sustained in the H-mode. The difference between the doses needed for a certain WCA is a factor of five or so. Good wettability is, therefore, obtained at lower doses of O-atoms in the case of plasma sustained in the E-mode. The possible explanation of this observation is taking into account the synergy between the atoms and other reactive species. The O-atoms should not heat the surface much because the intensity of heating due to heterogeneous surface recombination of O-atoms to parent molecules is low for polymeric materials [40]. On the other hand, plasma in the H-mode is a significant source of charged particles [41]. The positively charged oxygen ions cause significant etching of polymer materials [42] and thus the removal of polar surface functional groups that have been formed by the chemical interaction of O-atoms with the polymer surface. The etching is inhomogeneous, thus leading to nanostructuring of the polymer materials [43]. The nanostructuring progresses with the dose of ions, and, finally, a very rough surface of the nanoscale is obtained at large doses [44]. The very rich morphology on the nanoscale suppresses etching in the gaps, so the final surface finish is super-hydrophilic. Yet another possible explanation involves VUV radiation from oxygen plasma, but the science of interaction mechanisms between VUV photons and organic materials is still in its infancy.

The detailed explanation of the shift in the curve for H-mode towards larger fluencies, as evident from Figure 7, is beyond the scope of this paper. It is only possible to conclude that the O-atom dose needed for the super-hydrophilic surface finish of wheat seeds by using a mild plasma treatment in the E-mode is around 10^24^ m^−2^, while in the H-mode, it is several times larger, so the fluence close to 10^24^ m^−2^ will assure for the super-hydrophilic surface when plasma is in the H-mode.

The correlation between the surface functionalization, as probed by XPS, and wettability, as probed by WCA is interesting. The correlation is plotted in Figure 8. Again, the measured points are scattered, but the trend is obvious: the larger oxygen concentration on the surface of wheat seeds, as probed by XPS, will result in a lower WCA. The super-hydrophilic surface finish (immeasurably low WCA) is obtained at an oxygen concentration of over 30 at.%. For the E-mode treatment, the decrease in oxygen concentration is statistically significant for all WCA, except at ~50°. For the H-mode, the decrease in oxygen concentration is statistically significant for all the WCA. According to the results shown in Figure 2, the rest is carbon, with some impurities, the most significant being nitrogen. There is no reason for the functionalization of surfaces with nitrogen upon oxygen plasma treatment because the vacuum system in Figure 1 is hermetically tight, so no nitrogen is present in the processing gas. A feasible explanation for the increased concentration of nitrogen is the removal of the polyolefins and polysaccharides from the seed surface upon treatment with oxygen plasma. The subsurface is richer in proteins, so some nitrogen appears on the surface.

Here, it is worth mentioning that our initial WCA value of 110° is in good agreement with the findings of previous publications. Nearly all authors measuring WCAs of plasma-treated wheat seeds measured the untreated seed WCA in the range of 105–119°, while only one exception measured it at 92° [24]. The level of WCA decrease achieved by plasma treatment depended on the plasma system and treatment conditions. Bormashenko et al., who also used an IC RF plasma at low pressure, achieved super-hydrophilicity after 15 s of treatment [16]. Several other publications, using variations of atmospheric pressure DBD, mainly reduced the WCA to the range of about 40–60° [14,20,24,45], or in one case, only 87° [23], at the selected treatment conditions. Super-hydrophilicity was only achieved at much longer treatment times, such as 3 [19] or 15 min [15]. Treatment for 5 min with an RF plasma jet also decreased wheat seed WCA to about 74° [21]. These differences stem from the differences in reactive species density and homogeneity between the plasma systems. Low-pressure RF plasmas are known to rapidly hydrophilize organic surfaces, such as polymers [13], as well as the surface of seeds [27,32].

With plasma treatment of surfaces, hydrophilization is the result of both functionalization and etching [46]. As we have shown, our plasma treatment has achieved changes to both the surface chemistry and topography. The resulting effect is the exponential decrease of the WCA with increasing treatment time (Figure 5). A comparable evolution of the WCA was previously demonstrated using the same IC RF plasma system on seeds of beans [32] and alfalfa [31]. In wheat seeds, when WCA was investigated as a function of treatment time, its decrease was similarly exponential after DBD treatments using He [15], He and Ar [45], and air [19]. In other cases, a plateau was reached at a higher WCA [14,23].

### 3.4. Germination

Figure 9 shows the germination percentage of wheat seeds. Even for native, untreated seeds, the germination percentage was close to 100%. However, prolonged treatment times, i.e., 300 s in the E-mode and 10 s in the H-mode, showed a significantly reduced germination: a decrease of 15–20% was seen in the E-mode and a decrease of nearly 50% in the H-mode. This is explained by heating the seeds upon plasma treatment due to exothermic surface reactions. Unfortunately, the discharge system shown in Figure 1 does not allow for measuring the seed temperature during plasma treatment.

For commercial wheat seed batches, a high germination percentage is typically required by law. The germination percentage of our samples was near 100%. While in other work, plasma treatment has commonly been shown to stimulate seed germination of many agriculturally important species, this improvement has often been limited for wheat, at least in seed batches with high initial germination percentages, as summarized in our recent review [47]. One notable exception is the work of Bormashenko et al., where the germination percentage increased from about 47% to 83%. In other publications, plasma treatment under suitable conditions was able to stimulate wheat seed germination percentage by at least a few percent [15,24,45].

Our results indicate that at moderate exposure times, E and H-mode plasma treatments do not negatively affect wheat seed germination. Extensively long treatments, however, may still decrease the germination percentage. This was also previously demonstrated by Hoppanová et al., where air DBD treatments longer than 60 s notably decreased the natively 100% germination percentage [19]. In the works of Filatova et al., prolonged treatments using a CC RF air plasma also decreased previously improved germination percentages [7,48]. However, even if the germination percentage itself is unchanged, other parameters, such as seedling vigor, may be improved [19].

The changes seen at long treatment times in Figure 9 could be due to the heating of the seeds during the longer plasma exposure times. Typically, the temperature of seeds exposed to plasma will gradually increase. Using a coaxial DBD reactor, the measured temperature at the wheat seed coat barely exceeded 40 °C within 600 s of treatment, which did not negatively affect the germination percentage [45]. Elsewhere, wheat seed temperature increases of only a few °C were seen during 180 s of treatment with an air DBD [23] and 600 s of treatment with a nitrogen RF jet [18]. While heating in the E-mode is definitely not as extensive as in the H-mode, prolonged exposure to elevated temperatures could damage the seed embryo. Conversely, H-mode exposure results in extensive surface heating due to exothermal reactions of charged particles at the sample surface [49], but the thermal conductivity of organic matter is rather low, so it takes several seconds to heat the embryo beyond the tolerable temperature. This is a feasible explanation for the practically unchanged germination (Figure 9b) at treatment times up to about a second when using the H-mode; further treatment causes heat damage. Sound with the discussion about the densities of reactive plasma particles; the germination is suppressed at much shorter treatment times when using H-mode than E-mode from the treatment of wheat seed.

Alternatively, the negative effects on the germination percentage could also be the consequence of the etching of the surface, as shown in Figure 4, and the resulting damage to the seed coat. At prolonged treatment times (or, more precisely, the product of the treatment time and the RF power absorbed by plasma), the etching process may damage essential protective structures of the seed coat, which was previously shown for plasma-treated wheat seeds [15]. To the seed, the hydrophobic properties of the cuticle offer a protective barrier that regulates the seed coat permeability and offers protection against stressors [50]. Etching may thereby remove a sufficiently thick layer to make the seed surface vulnerable to pathogens, such as fungi. The pathogen contamination, in turn, decreases the percentage of healthy seeds and thereby the germination percentage.

### 3.5. Viability of Seeds

Seed viability was checked by colorimetric assay using 0.1% tetrazolium chloride. The tetrazolium chloride test is complementary to the more elaborate germination percentage determination. Figure 10a shows the viability of seeds treated in the E and H-modes at different treatment times. The measured values somewhat correlate with the germination of seeds at these treatment times (Figure 9). The viability is reduced by about 10% and 20% in the E-mode after 100 and 300 s of exposure time, respectively. As expected, the viability was only about 50% of the original value in the H-mode after 10 s of exposure to plasma. The difference in viability is statistically significant for 100 and 300 s of treatment in E-mode and 3 and 10 s in H-mode. Figure 10b shows the viability versus the product of the real power absorbed by plasma and treatment time. The viability is not affected by about 100 Ws. After that, the viability is significantly lower compared to control at 3000 and 10,000 Ws for treatment in E-mode and ~1000 and 3000 Ws for treatment in H-mode. Taking into account the results presented in Figure 6, it is possible to conclude that the super-hydrophilic surface finish always causes a significant loss in viability. Namely, Figure 6 indicates that WCA is around 70° as the product of the real power and treatment time of 100 Ws. This observation indicates a definite drawback in any attempt to use oxygen-plasma technology in agricultural practice. The results of the viability assay are shown in Figure 10c. Seeds turned red if viable and stayed uncolored (yellow) if non-viable.

## 4. Conclusions

Wheat seeds of the cultivar Bologna were treated in low-pressure RF oxygen plasma in the E and H-modes. Oxygen plasma made the surface hydrophilic by incorporating oxygen functional groups on the surface. This was confirmed by the surface elemental composition of plasma-treated seeds, which shows that the C/O ratio decreased, meaning that at.% of C on the outermost surface layer decreased while the at.% of O increased. The required oxygen concentration in the surface film to obtain the super-hydrophilic surface finish was found to be over 35 at.%. The environmental SEM micrographs of wheat seeds showed interesting morphological features on the surface of wheat seeds. The surface of the untreated seed was uneven, with long linear groves stretching across the surface area. When exposed to plasma in E-mode, these grooves became almost flat, while in H-mode plasma, cracks appeared, and the surface was heavily eroded. Consequently, to both functionalization and etching, the WCA dropped from an initial 110° to 10° after 100 s in E-mode, while after 300 s, the surface became super-hydrophilic. In H-mode, however, the hydrophilic effect was achieved much faster. Super-hydrophilicity was already achieved after 10 s of treatment in H-mode. Yet, achieving this level of surface hydrophilicity was incompatible with the successful germination of wheat seeds. Prolonged exposure time to plasma lowered the germination percentage by 20% in E-mode and almost 50% in H-mode, which was the consequence of extensive etching of the outer seed coat layers, as well as possibly heating of the seeds, during the prolonged exposure times in plasma. This also contributed to lowered viability of seeds treated under the same conditions. All of the above results demonstrate that low-pressure IC RF oxygen plasma is a powerful tool for surface wettability modification of wheat seeds and that at suitable treatment conditions, it does not negatively affect the germination percentage.

## Figures and Tables

**Figure 1 plants-11-01552-f001:**
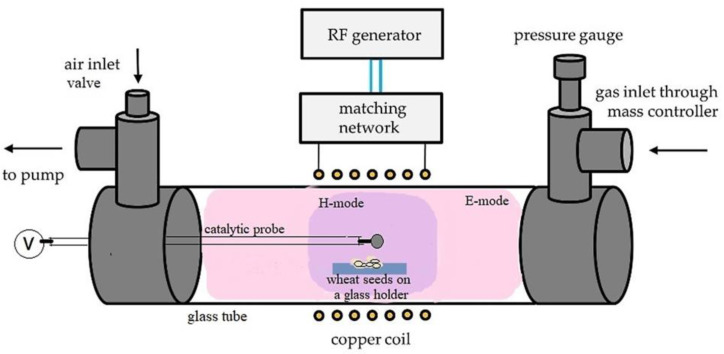
Schematic representation of the plasma system used for the treatment of wheat seeds. Adapted from [27].

**Figure 2 plants-11-01552-f002:**
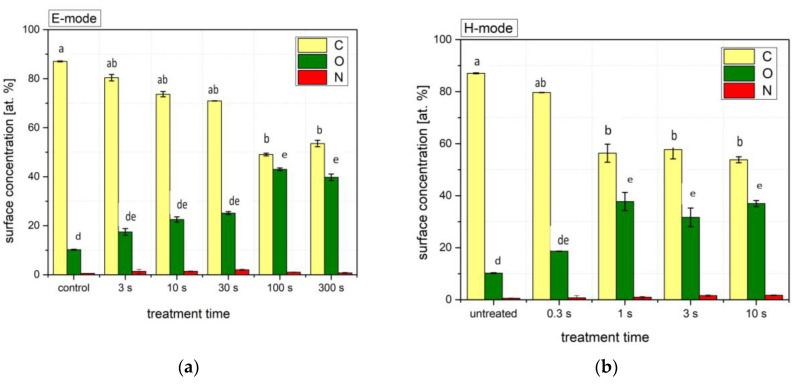
The elemental concentration of wheat seed surface of carbon, oxygen, and nitrogen determined by XPS after treatment with (**a**) E-mode or (**b**) H-mode plasma. The error bars represent standard error. Different lowercase letters above the bars represent statistically significant differences (*p* < 0.05; post hoc Tukey’s range tests) between treatments for each individual element (C or O). Mean values (±SE) for the respective duplicates are given.

**Figure 3 plants-11-01552-f003:**
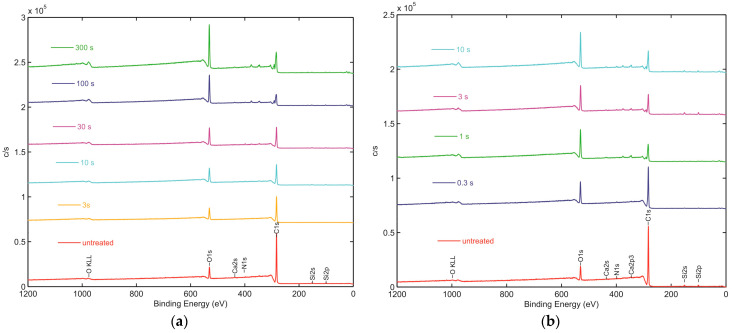
XPS survey spectra of untreated and plasma-treated wheat seeds in the (**a**) E-mode and (**b**) H-mode.

**Figure 4 plants-11-01552-f004:**
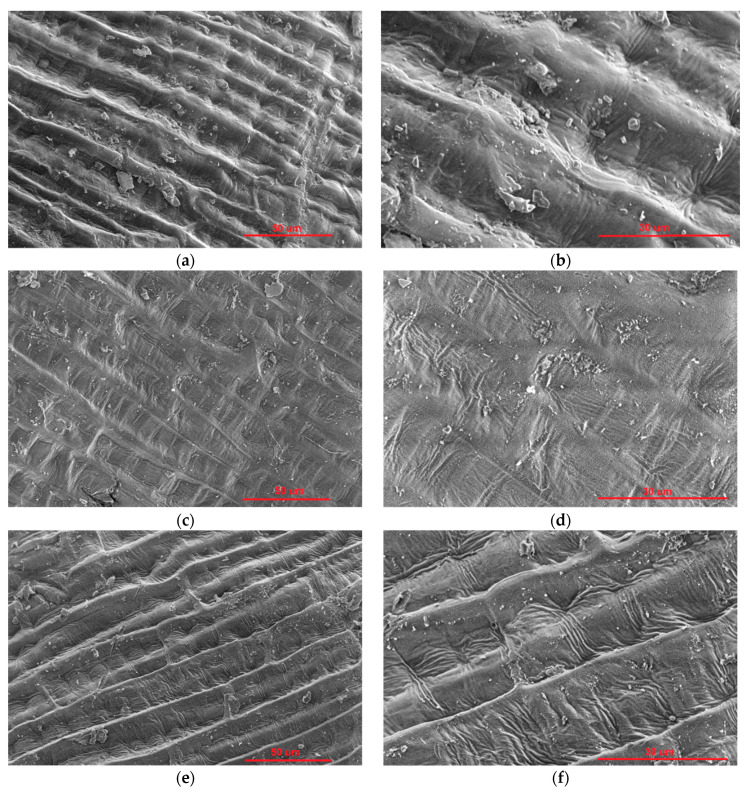
SEM images of Bologna wheat surface: (**a**,**b**) untreated; (**c**,**d**) E-mode treatment, 300 s; (**e**,**f**) H-mode treatment, 10 s. (**a**,**c**,**e**) 2000× magnification; (**b**,**d**,**f**) 5000× magnification.

**Figure 5 plants-11-01552-f005:**
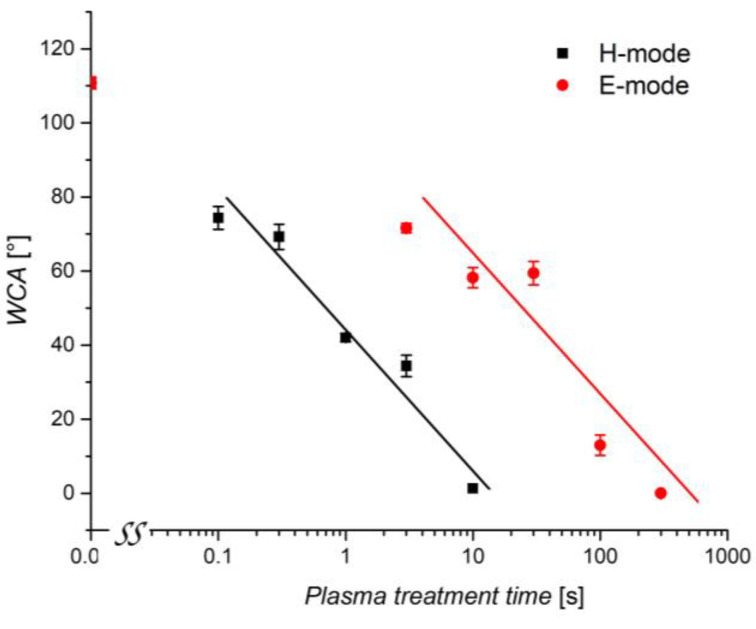
Water contact angle (WCA) on the wheat seed surface after treatment with an E-mode or H-mode plasma at various treatment times. The error bars represent standard error. Mean values (±SE) for the respective triplicates of WCA are given.

**Figure 6 plants-11-01552-f006:**
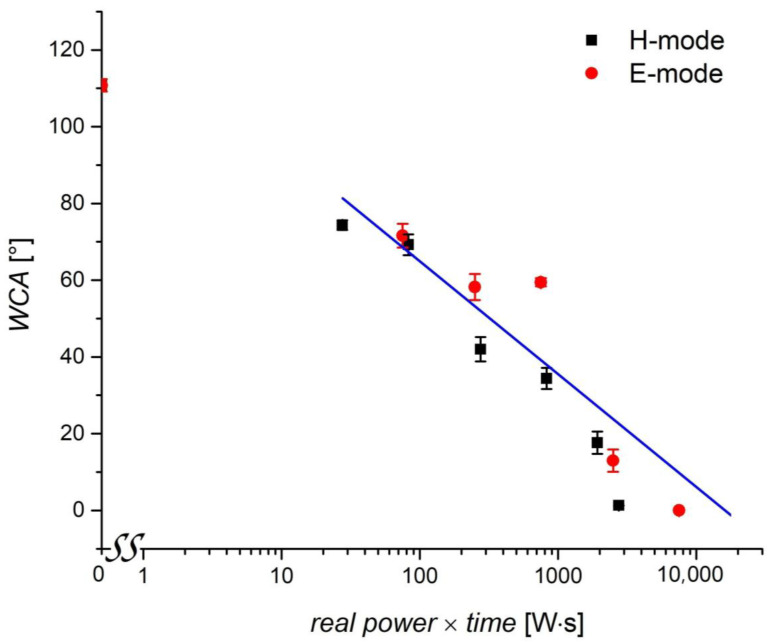
The evolution of surface wettability versus the product of the discharge power absorbed by the plasma. The error bars represent standard error. Mean values (±SE) for the respective triplicates of WCA are given.

**Figure 7 plants-11-01552-f007:**
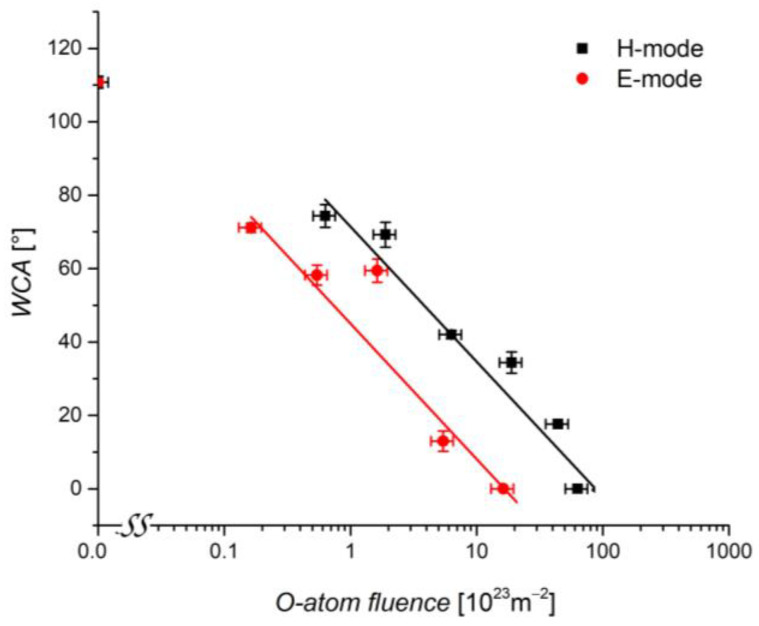
Water contact angle (WCA) on the wheat seed surface versus the fluence of O-atoms in E-mode or H-mode plasma at various treatment times. The error bars represent standard error. Mean values (±SE) for the respective triplicates of WCA are given.

**Figure 8 plants-11-01552-f008:**
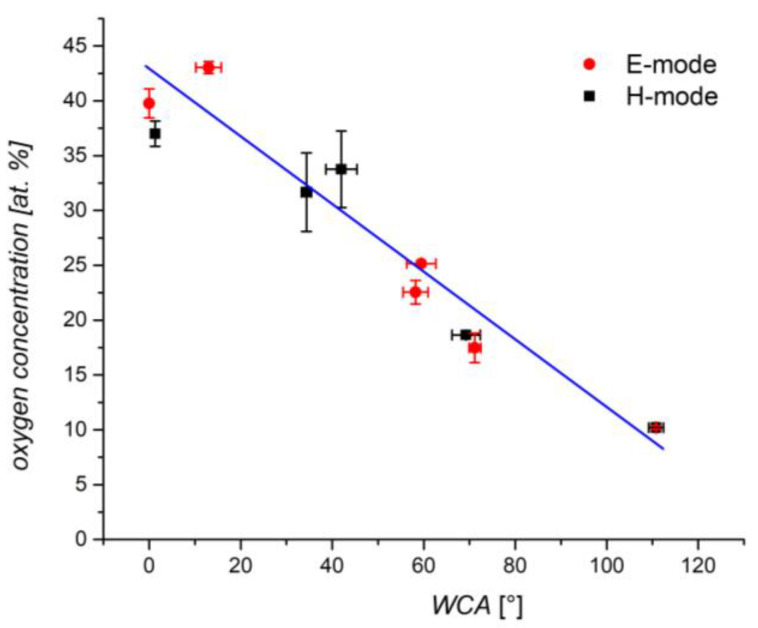
The oxygen concentration versus the water contact angle after treatment in E and H-mode’s plasma. The error bars represent standard error. Mean values (±SE) for the respective triplicates of WCA are given.

**Figure 9 plants-11-01552-f009:**
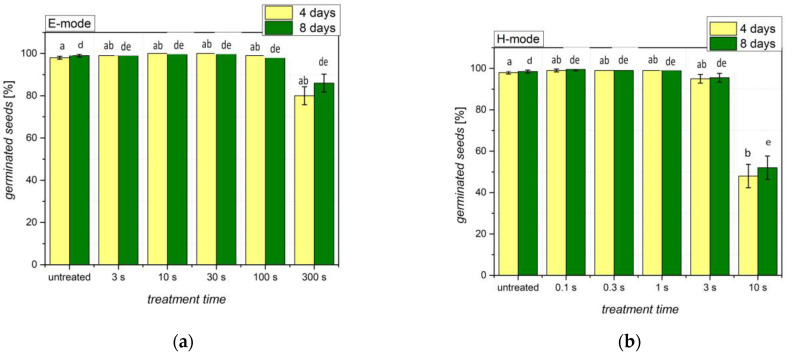
Germination percentage of untreated and plasma-treated wheat seeds after 4 and 8 days of incubation at germination conditions. Wheat seeds are treated in the (**a**) E-mode and (**b**) H-mode. The error bars represent the standard error. Different lowercase letters above the bars represent statistically significant differences (*p* < 0.05; post hoc Tukey’s range tests) between treatments for an incubation time of 4 and 8 days. Mean values (±SE) for the respective duplicates are given.

**Figure 10 plants-11-01552-f010:**
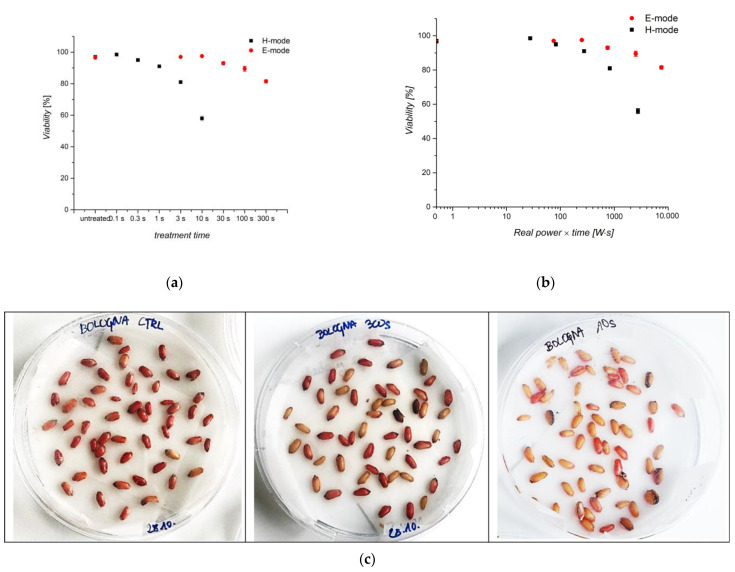
(**a**) Comparison of seed viability after treatment in E and H-mode plasma at different treatment times. (**b**) Correlation between seed viability versus real power and treatment time in plasma. (**c**) Results of the colorimetric assay with tetrazolium chloride. From left to right: untreated seeds; 300 s treatment in E-mode; 10 s treatment in H-mode. The error bars in (**a**,**b**) represent standard error mean values (±SE) for the respective duplicates given.

## Data Availability

Not applicable.

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
