# Peer review of "Surface Modifications of Wheat Cultivar Bologna upon Treatment with Non-Equilibrium Gaseous Plasma"

_plants, 2022, doi:10.3390/plants11121552_

Round 1
Reviewer 1 Report
The Authors study an important and actual modality to increase the germination, An increasingly important challenge for the world community even in conditions such as those caused by climate change. They have examined many aspects of these "new" techniques deeply.
The introduction provides an exhaustive and well-organized picture of the state of the art of the topic they have decided to address. The experimental part is wide, covers various aspects of the object of study and is carried out in a rigorous manner. The graphical representations of the results obtained are clear and easy to understand. The results are well discussed and offer various insights and subsequent possible technological evolution for a possible use and dissemination of these techniques.
In fact the conclusion are well argued and useful for those who work in the same field.
The bibliography is updated and corresponding to what is reported in the manuscript.
I think the paper deserves the publication.
Author Response
"Please see the attachment."

Reviewer 2 Report
The paper presents an experimental study of wheat cultivar treated by low-pressure inductively coupled plasma. The RF oxygen discharge is operated in E and H mode for powers of 25 and 275 W with treatment times between 0.1 and 300 s. Authors observed the modification of surface, after the plasma treatment, employing XPS and SEM. It was observed that functionalization/etching of the surface modified the surface wettability. The authors obtained high germination rate after samples treatment by plasma while the seed viability remained unaffected until about 100 W.s. Long treatments lead to the decrease of germination rate and viability. The paper fits the scope of the Plants Journal and may be accepted for publication after authors address to the following questions:
- WCA in the abstract? Please used water contact angle.
- In the introduction authors put in relief the importance of low-temperature plasmas for crops treatment. It can be cited an important and recent reference in this area: J. Loureiro and J. Amorim, Kinetics and Spectroscopy of Low Temperature Plasmas, DOI 10.1007/978-3-319- 09253-9 when discussing this topic.
- In the introduction section state what is the new contribution of this work related to the others found in the literature.
- Please insert in the materials and methods section a description of the RF generator used, the matching network etc. The plasma discharge is very different in these two operating modes E and H. Explain how theses modes are characterized in your experiments.
- The purity of the gas used in the experiments is an important issue. Impurities may lead to the production of radicals and influence the discharge and the seeds treated. Please give the purity of the oxygen gas employed in the experiments.
- Please give the uncertainty of oxygen atoms measurements in the H mode of the discharge.
- Authors attribute to the oxygen atoms the main role in the surface modification. What can be said about the singlet metastable state ?2(? 1Δg) which may have high density in these discharges? Please comment and update the the manuscript.
- Improve the quality of figure 3.
- In XPS spectra after plasma treatment peaks related to N and Ca appears. What is the reason?
- How long stands the wettability of the sample treated?
- What is the more efficient discharge mode for seeds treatments considering the parameters evaluated in this work?
Author Response
"Please see the attachment."

Reviewer 3 Report
The study “ Surface modifications of wheat cultivar Bologna upon treatment with non-equilibrium gaseous plasma” is one of numerous studies published on this topic. Unfortunately, I do not see any novelty in this work, because the effects of cold plasma on the surfaces of various seeds and grains are relatively well studied and described. Also the findings that long exposure to cold plasma reduces the germination and vitality of seeds/grains, the vitality of young seedlings and negatively affects the subsequent growth and development are well described in previously published literature.
In addition, the quality of manuscript preparation, presentation and evaluation of results, discussion and formulation of sentences must be substantially improved. In general, the presentation of study can be characterized as careless.
Remarks and suggestions for improvement:
The introduction is rather messy and has to be rewritten, particularly the final part where you have to clearly formulate the aims of the work.
The graphical evaluation of the results and legends are really chaotic. In the graphs, the H- and E modes are shown in color (red and black), in the legend and in the text, the authors refer to them as a and b. E.g. in the case of Figure 7, the legend also mentions c, which I don't understand at all, in Figure 8, a and b is missing in the legend. In line 435, the authors refer to Figure 5 (d), that's what?
Statistical data processing is quite strange. In Figure 5, 6, 7, the authors in the legend claim that values are statistically significant compared to control at p < 0.001. This is an interesting statement, but nothing supports it as only error bars (probably SD) are shown in the graphs.
I absolutely do not understand the statistical processing of data in Figure 2. For example, the elemental concentration of wheat seed surface of carbon decreased in all variants (both in E and H mode) compared to the control. How is it possible that e.g. variants of 30 s and 100 s are statistically significant different at p < 0.001 and variant 300 s, which has a higher average value than 100 s and a lower average value than 30 s is statistically significant different at p < 0.01. This is complete nonsense.
Overall, the number of repetitions of experiments is missing in the Material and methods as well as in the legends of the Figures. Also, Standard deviation (SD) more accurately reflects the real state than Standard error (SE).
In Figure 4, it would be appropriate to use the arrows to indicate what is important.
Why there is no statistical evaluation of the data in the Figure 9?
There are a large number of errors in the text (eg citing authors instead of using numbers) and in the list of cited literature. You should look at the instructions for the authors.
Numerous similar remarks, indicating insufficient quality of the submitted manuscript, can be given for the rest part of the manuscript. However, the arguments mentioned above are sufficient to substantiate the suggestion to reject the submitted paper.
Author Response
"Please see the attachment."

Round 2
Reviewer 2 Report
The paper may be accepted for publication in this revised version.
Author Response
Thank you very much for the positive opinion.
Reviewer 3 Report
Thank you for accepting the comments and recommendations
Author Response
Thank you for your positive reply.